# Impact of the Covid-19 pandemic on inpatient health care in Switzerland 2020–2021—A descriptive retrospective study using admission data of all Swiss hospitals

Brigitte Wirth[1], Michael Stucki[2], Reto Joerg[3], Christoph Thommen[2], Marc Höglinger[2]*

1 Integrative Spinal Research Group, Department of Chiropractic Medicine, Balgrist University Hospital and University of Zurich, Zurich, Switzerland, 2 Winterthur Institute of Health Economics, School of Management and Law, Zurich University of Applied Sciences, Winterthur, Switzerland, 3 Swiss Health Observatory, Neuchâtel, Switzerland

* marc.hoeglinger@zhaw.ch

## Abstract

### Background

As part of the Covid-19-restrictions in Switzerland, a federal ban on non-urgent examinations and treatments was applied to all hospitals during six weeks in spring 2020 ("spring lockdown"). The aim of this study was to comprehensively investigate the consequences of the Covid-19 pandemic on Swiss inpatient admissions based on data of *all* hospitals, focusing on selected procedures of different medical urgency.

### Methods

The study includes all acute care inpatient cases (including Covid-19 cases, excluding cases in psychiatry and rehabilitation) according to the Swiss Medical Statistics of Hospitals. Besides the total number of admissions, subdivided by regions, hospital types and age groups, we focused on selected procedures representing different medical urgency: elective surgeries, cancer surgeries, and emergencies. Procedures were selected based on expert interviews. We compared the number of admissions during spring lockdown and for the whole years 2020 and 2021 in absolute numbers and in percentage changes to the corresponding periods in 2019 (baseline year).

### Results

During spring lockdown, the number of admissions decreased by 47,156 (32.2%) without catch-up effect by the end of 2020 (-72,817 admissions/-5.8%). With procedure-specific decreases of up to 86%, the decline in admissions was largest for elective surgery, a decline that was only fully reversed in the case of a few procedures, such as joint arthroplasty. Strikingly, admissions due to emergencies also substantially decreased during spring lockdown (stroke -14%; acute myocardial infarction STEMI: -9%, NSTEMI: -26%). Results for the

**Data Availability Statement:** The data that support the findings of this study are available from the Swiss Federal Statistical Office (SFSO), Espace de

l'Europe 10, CH-2010 Neuchâtel, Switzerland.
Restrictions apply to the availability of these data,
which were used under license for the current
study, and so are not publicly available. Requests
can be sent directly to the SFSO via email at
Gesundheit_DSV@bfs.admin.ch.

**Funding:** The study received funding from the
Swiss Health Observatory (OBSAN). One of the co-
authors, Reto Joerg (RJ), is an employee of
OBSAN. RJ contributed to the study's design, data
collection, analysis, and manuscript preparation.
OBSAN is a Swiss competence, service, and
information center, endorsed by the Confederation
and the cantons. It focuses on conducting scientific
analyses and providing information concerning the
population's health, the healthcare system, and
health policy.

**Competing interests:** The authors declare that they
have no competing interests.

selected procedures in cancer surgery showed no consistent pattern. In 2021, admission
numbers for most procedures reached or even exceeded those in 2019.

## Conclusions

The substantial reduction in admissions, particularly in elective procedures, may reflect the
impact of the triage in favor of anticipated Covid-19-cases during spring lockdown. By the
end of 2020, admissions were still at lower levels relative to the previous, pre-pandemic
year. The numbers in 2021 reached the same levels as those in 2019, which suggests that
the Covid-19 pandemic only temporarily impacted inpatient health care in Switzerland.
Long-term consequences of the observed reduction in admissions for emergencies and
cancer surgery need to be investigated at the individual level.

## Introduction

At year-end 2019, the Covid-19 virus spread worldwide and was declared a pandemic by the
WHO in March 2020 [1]. The first Covid-19 cases in Switzerland were registered in February
2020 [2]. Although restrictions in Switzerland were comparatively mild [3], the Covid-19 pan-
demic put the Swiss healthcare system to the test in various respects. On the one hand, many
Covid-19 patients in need of intensive care had to be treated within a short period of time. On
the other hand, treatments not related to Covid-19 were reduced to guarantee adequate care
for increasing actual and anticipated numbers of Covid-19 patients. Due to this extraordinary
situation, a federal ban on "medically non-urgent examinations and treatments" applied to
hospitals throughout Switzerland between March 16 and April 26, 2020 ("spring lockdown")
[2]. After this period, although no regulatory measures were put into effect, the federal authori-
ties repeatedly asked hospitals to reserve inpatient capacities for a possible rise in Covid-19
patients. Temporarily, case numbers in intensive care units peaked due to an increase in
Covid-19 cases. Hospitals therefore postponed interventions on a voluntary basis even after
the spring lockdown, e.g., because of higher needs for intensive care units [4], particularly in
autumn/winter 2020, when the number of infections increased again. In addition to these
restrictions on the supply side, patients' demand for medical services may have declined due to
fears of a Covid-19 infection or to self-imposed restraints to avoid additional burden on the
strained healthcare system. Evidence for these demand-side effects from patients for the period
of the spring lockdown and subsequent months can be found in a representative Swiss survey
study [5].

The Covid-19 pandemic provides an unwanted, but unique, opportunity to investigate a
healthcare system's resilience, i.e. the ability to respond effectively to shocks [6, 7] and to com-
pare this ability between health care systems. In a synthesis of global studies, a recent system-
atic review reported large reductions in the utilization of all types of health care services
(outpatient: median -42%; diagnostic and imaging procedures: -31%; therapeutic and preven-
tive care: -30%; and hospital admissions: -28%) in pandemic versus non-pandemic periods [8].
Interestingly, this applies not only to elective surgery, e.g. knee and hip arthroplasty [9–11],
but also to emergencies such as stroke and acute myocardial infarction [10, 12–18]. Although
the reasons for this phenomenon are not clear yet, besides fear of infection, also reductions in
pollution [19] and stress [15] or reduced transmission of other infections that trigger cardio-
vascular diseases (due to social distancing) [18] are discussed as potential causes in scientific

literature. Cancer surgery was also cut back during the restrictions due to Covid-19, depending on the extent of restrictions: a prospective cohort study on planned surgery for 15 tumor types in 61 countries reported non-operation rates of 0.6% in light restrictions, 5.5% in moderate, and 15% in full lockdowns (median follow-up of 23 weeks) [20]. Based on numbers from England, it was estimated that for incident cancers, delays in surgery of three and six months were associated with reductions in life-years gained of 19% and 43%, respectively [21].

Switzerland, with a population of approx. 8.7m inhabitants by the end of 2020 [22], has a healthcare system of high quality at high costs: its Healthcare Access and Quality Index (HAQ Index) is the third highest in the world [23], and its annual healthcare spending the second highest of all OECD countries [24]. According to previous estimates, the impact of the Covid-19 pandemic on the Swiss healthcare system was comparable to other countries [8]: hospital admissions decreased during the spring lockdown by around 30%, depending on data source: 27% (8,580 admissions) in a study using data of a single health insurer (insuring 16% of the Swiss population) [25], and 34% based on data of 38 Swiss hospitals (clearing around 35% of hospital admissions; approx. 237,500 admissions in the first half of 2020) [26]. The reductions in admissions were clear in both elective and less elective medical fields [26]. A recent Swiss study relying on claims data of another health insurer (insuring 6% of the Swiss population) focused on orthopedics and cardiovascular procedures. The authors estimated non-pandemic weekly procedure numbers based on control years and distinguished elective from emergency admissions [27]. For elective admissions, they reported a reduction of 53% for orthopedic and of 46% for cardiovascular procedures during spring lockdown. By the end of 2020, the reduction was 10% for elective orthopedic and 8% for elective cardiovascular procedures. For emergency admissions, the reductions were 17% for orthopedic and 15% for cardiovascular procedures during spring lockdown. By the end of 2020, orthopedic emergency admissions were 3% and cardiovascular emergency admissions 2% lower than estimated [27].

However, these studies [25–27] are based on subsets of the Swiss population and might thus under- or overestimate the *real* Covid-19-related changes in the provision of inpatient care. Furthermore, although the differentiation between elective and emergency procedures is a first step, a more detailed insight into the extent to which particular procedures were affected is needed [27]. Thus, the aim of the present study was to investigate the consequences of the Covid-19 pandemic on inpatient health care in Switzerland based on data of *all* Swiss hospitals in the years 2019–2021, focusing on selected treatments representing procedures of different medical urgency: elective surgery, cancer surgery, and emergencies.

## Methods

### Data

The study is based on data of the Swiss Medical Statistics of Hospitals (MS) and considers all admissions in acute-somatic inpatient care (including Covid-19-cases, excluding cases in psychiatry and rehabilitation). MS Data contains a main diagnosis for each hospitalization and up to 49 secondary diagnoses according to the International Classification of Diseases, German modification, version 10 (ICD-10-GM) as well as one main and up to 100 secondary treatments according to the Swiss inpatient procedure codes (CHOP). For this study, we only used the main diagnosis and the main treatment codes for each case. We thereby avoided double-counting cases in the subsequent classification of cases into specific procedures. Cases were assigned to a calendar week according to their date of admission. As MS data only contains cases with known date of admission and discharge, the data for the most recent year (2021) did not contain all admissions in that year. To ensure comparability over the years, inpatient stays stretching over year-end in any year (2016–2021) were excluded from the analyses. As

the latter concerns only about 1% of all admissions each year and applies to all years during the period studied, the potential impacts on the results are negligible.

This study does not use individual human subject data, but solely administrative, aggregated MS data on hospital stays that are routinely collected by the Swiss Federal Statistical Office (SFSO). Data collection is regulated by law and conforms to national ethics and data protection regulation. Data were accessed for research purposes on March 16, 2022. As the data is aggregated and anonymized an individual consent is not necessary (not even possible). Consequently, authors had no access to information that could identify individual participants during or after data collection. This is confirmed by the ethic committee of the cantonal department of health (Kantonale Ethikkommission des Kantons Zürich), which approved the study and concluded that it did not fall under the Swiss Human Research Act (Reference-Number: BASEC-Nr. Req-2022-01580). All methods were carried out in accordance with relevant guidelines and regulations in the declaration of Helsinki.

## Analyses

We analyzed changes in inpatient admissions due to the Covid-19 pandemic based on (i) the number of total admissions, (ii) the number of total admissions by Swiss regions (Lake Geneva Region, Espace Mittelland, Northwestern Switzerland, Zurich, Eastern Switzerland, Central Switzerland, Ticino [28]), type of hospital (university hospital, major hospitals but not affiliated to university, small regional hospitals, and specialized clinics, e.g., special surgical clinics, pediatric, and maternity hospitals [29]), patient age (0–17 years, 18–39, 40–64, 65–79, 80+), and (iii) data for 13 selected procedures representing different levels of medical urgency: elective surgery (N = 5), cancer surgery (N = 4; the three most frequent cancer types for men and women, respectively [30]), and emergencies (N = 4). These procedures were determined in interviews with seven experts. The selection criteria were a certain intra-category homogeneity regarding medical urgency, a minimal caseload, and practical relevance according to international literature (particularly regarding the "missing" strokes and heart attacks) [9, 15, 21, 31]. Furthermore, the procedures must not be shiftable from inpatient care to the outpatient sector to avoid potential biases because the outpatient sector is not covered in the data. The selected procedures and the defining ICD- and CHOP-codes are presented in Table 1. We compared admission numbers from 2020 and 2021 with those from 2019. Because inpatient care in a particular field can be subject to considerable systematic changes from one year to another, the meaningfulness of the comparison decreases with each year further back in time. Because changes might occur abruptly, statistical modeling using linear and non-linear trends is limited and rests on questionable ceteris paribus assumptions. 2019, the year prior to the Covid-19 pandemic, thus corresponds to the best possible control period, since it can be assumed that disease prevalence, hospital infrastructure, medical coding guidelines and classification systems, reimbursement schemes, as well as other potentially relevant factors change little within a year.

Similarly, other trends, including population aging, likely had minimal impact on our findings, given the limited timeframe of our analysis. This is evidenced by the minimal yearly variation in total case numbers from 2016 to 2019, with annual changes ranging from -0.1% to +0.2%, compared to a 5.8% decrease in 2020. We calculated absolute case counts and relative changes compared to the baseline year 2019 during the weeks of the spring lockdown 2020 (calendar weeks 12–17), when the Oxford stringency index combining nine indicators of Covid-19-measures was at its maximum for Switzerland [3].We also analyzed cases for the full year 2020, as well as 2021 (calendar weeks 2–52; start is week 2 because this was the first complete week, i.e., the first week that consisted of seven days in the years 2019 and 2020). We

**Table 1. ICD- and CHOP-codes of selected procedures.**

| Procedure | ICD-10-GM*) | CHOP*) | |
|---|---|---|---|
| **Elective surgery** | | | |
| Knee prosthesis (incl. partial prosthesis) for knee OA | M17 | 81.54.2.- | |
| Hip prosthesis (incl. partial prosthesis) for hip OA | M16 | 81.51.1.- | |
| | | 81.52.2.- | |
| | | 81.52.3.- | |
| Reconstruction of the anterior cruciate ligament | M23.61 | 81.45.- | |
| | S83.53 | 81.95.10 | |
| | | 81.95.11 | |
| Reconstruction of hallux valgus | M20.1 | 77.51 | |
| | | 77.52 | |
| | | 77.53 | |
| Transurethral prostatectomy (TUR-P) | | 60.2.- | |
| **Cancer surgery** | | | |
| Mastectomy in breast cancer | C50.- | 85.26 | up to and including 2019 |
| | | 85.34.- | up to and including 2019 |
| | | 85.4.- | up to and including 2019 |
| | | 85.A0 | from 2020 |
| | | 85.A1 | from 2020 |
| | | 85.A2 | from 2020 |
| | | 85.A3 | from 2020 |
| | | 85.A4 | from 2020 |
| | | 85.A5 | from 2020 |
| | | 85.A6 | from 2020 |
| | | 85.A7 | from 2020 |
| Prostatectomy in prostate cancer | C61 | 60.5.- | |
| | | 60.3 | |
| | | 60.4 | |
| | | 60.99.11 | |
| Lobectomy or pneumonectomy in lung cancer | C34.- | 32.4.- | |
| | | 32.5.- | |
| Colorectal resection in colorectal cancer | C18.- | 45.7.- | |
| | C19 | | |
| | C20 | | |
| **Emergencies** | | | |
| Stroke | I63 | | |
| Acute myocardial infarction NSTEMI | I21.4 | | |
| Acute myocardial infarction STEMI | I21.0 | | |
| | I21.1 | | |
| | I21.2 | | |
| | I21.3 | | |
| Appendectomy in acute appendicitis | K35 | 47.00 | |
| | | 47.01 | |
| | | 47.02 | |
| | | 47.09 | |

*) Only cases in which the diagnosis was coded as the main diagnosis and the treatment as the principal treatment

CHOP = Coding Swiss Classification of Operations; ICD = International Statistical Classification of Diseases and Related Health Problems, German Modification;

OA = Osteoarthritis

report Incidence Rate Ratios (IRRs) and corresponding 95% confidence intervals derived from a univariable Poisson regression when analyzing selected groups of procedures, some of which had low case numbers. The IRRs compare incidence rates between two time points (2020 and 2021 vs. 2019), calculated as $IRR = \frac{inpatient\ stays_{2020/2021}}{inpatient\ stays_{2019}}$. An IRR of 1 indicates no difference in incidence rates between the two periods. We assume a constant population size and characteristics over the comparison period. The IRR is obtained by exponentiating the regression coefficient for the comparison year. For background on Poisson regression, see reference [32], and for a recent application to hospital admissions, see reference [33]. This method allows us to address estimation uncertainty, especially pertinent in our analysis of the selected procedures with low case numbers. All analyses were made using Stata version 17.

## Results

### Total number of admissions

During the spring lockdown (calendar weeks 12–17), the decrease in admissions amounted to 47,156, or 32.2%, compared to the previous year (99,378 in 2020 compared to 146,534 in 2019) (Fig 1). Regarding the entire year 2020, 1,188,700 admissions were registered, which results in a reduction of 72,817 or 5.8% compared to the previous year's 1,261,517 admissions. Thus, the reduction during the spring lockdown was not compensated for (i.e., the number of admissions did not reach the pre-pandemic level) by the end of 2020. In fact, the difference compared to the previous year increased even further by a factor of 1.5 (from 47,156 during the spring lockdown to 72,817 by the end of 2020). This is particularly noticeable in autumn/

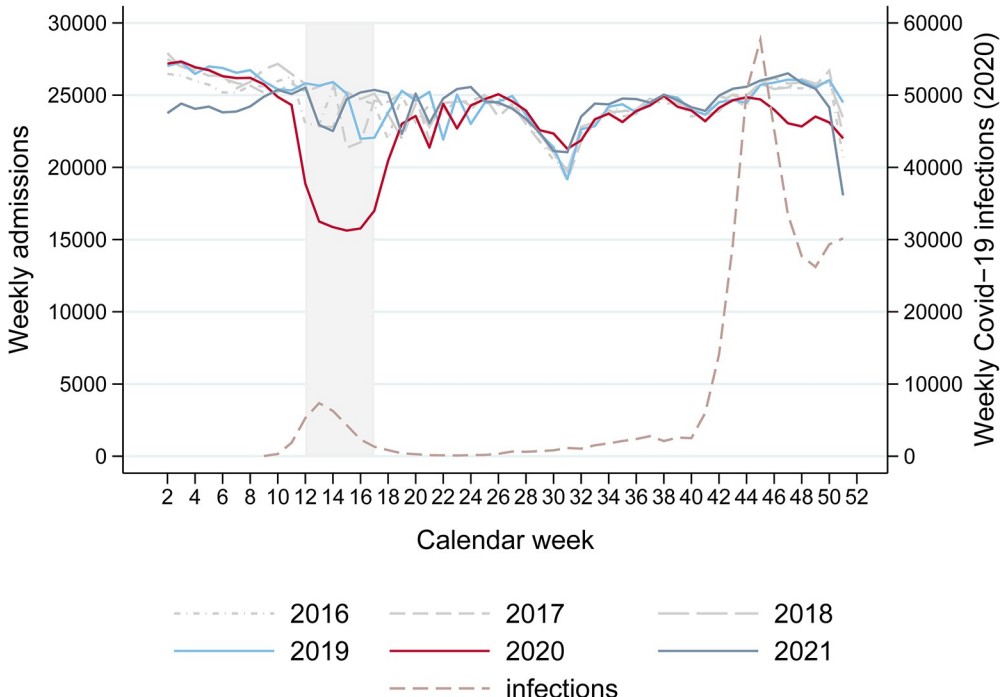

**Fig 1. Weekly numbers of total admissions by years 2016–2021 and of Covid-19 cases in 2020.** The graph shows the weekly numbers of total admissions in the years 2016–2021 and the numbers of Covid-19 cases in 2020 [3] during the calendar weeks 2–51. Start is week 2 because this was the first complete week, i.e., the first week that consisted of seven days in the years 2019 and 2020. Week 52 is not displayed as surgery activity in this week highly depends on when the holidays were within this week. The shaded area indicates the lockdown (calendar weeks 12–17).

winter 2020, from calendar week 41 onwards, when Covid-19 case-numbers increased rapidly again (second wave) [34]. In 2021, the total number of admissions amounted to 1,249,548, which is 0.9% less than in 2019. These numbers include 36,284 Covid-19 cases in 2020 and 35,628 in 2021 (3.1%/2.9% of all admissions; ICD code U07.1: cases in which Covid-19 has been detected by a laboratory test, or U07.2: non laboratory confirmed Covid-19 cases).

## Admissions by region, type of hospital and patient age

Looking at the number of admissions by Swiss major regions [28], the patterns are consistent: a sizeable drop in admissions during the spring lockdown 2020 (Fig 2, A1) and a reduction of roughly 4 to 6% by the end of 2020 (Fig 2, A2). Strikingly, the admission numbers in Ticino followed the general pattern but were already significantly below the previous year's numbers before the spring lockdown, and the decline was with -10% more pronounced than in the other regions. The decrease during the spring lockdown was highest for specialized clinics, e.g., special surgical clinics, pediatric, and maternity hospitals (Fig 2, B1) [29]. However, admission numbers in specialized clinics partly caught up right after the lockdown and by the end of the year they showed a relatively small decrease of -3.3% compared to 2019. University hospitals, in contrast, experienced another drop in autumn/winter 2020 (week 41 to year-end), resulting in the largest reduction of -7.4% by the end of the year (Fig 2, B2). As for patients' age, the reduction during the spring lockdown was most pronounced for patients aged 40–64 and 65–79 (Fig 2, C1). These numbers, however, caught up by year-end, while admissions of children and patients 80+ showed a smaller decline during the spring lockdown, but the catch-up effect was smaller (Fig 2, C2).

## Selected procedures: Elective and cancer surgeries, emergencies

During spring lockdown, the number of the selected *elective surgeries* dropped by between 71.4% (transurethral prostatectomy, TUR-P) and 86.4% (total knee replacement and hallux valgus reconstruction) (Fig 3). Remarkably, the decline in surgeries for total joint arthroplasty and for TUR-P was largely reversed after the spring lockdown (decrease in absolute differences in admission numbers relative to 2019 between spring lockdown and year-end 2020). This resulted in a marginal all-year reduction of 1% for hip and knee replacement and 4% for TUR-P when comparing the whole of 2020 to 2019. In 2021, admission numbers for these procedures were higher than in 2019 (Fig 3, Table 2). In contrast, for the other elective procedures considered, it was not possible to reverse the decline in admissions during the spring lockdown 2020, and these actually increased until year-end (increase in relative difference in absolute numbers between post-spring lockdown and year-end 2020). This resulted in an all-year reduction of 12% for hallux valgus reconstruction and of 20% for the reconstruction of the anterior cruciate ligament (ACL) by the end of 2020. Even in 2021, these procedures showed no return to pre-pandemic levels, with admission numbers 15.3% and 11.6% respectively lower than before the pandemic.

In the selected *cancer surgeries*, changes in admissions showed a less consistent pattern and incidence rate ratio (IRR) estimates were less precise due to the low case numbers. During spring lockdown, admissions decreased considerably for colorectal resection in colorectal cancer (-20%). Lobectomy/pneumonectomy in lung cancer (-13%), and mastectomy in breast cancer (-5%) showed a smaller decrease and an IRR that is not statistically significantly different from 1. By the end of 2020, the numbers of colorectal resections and mastectomies were approximately the same as in 2019, while the number of lobectomies/pneumonectomies was still lower (-12%). Similarly, by the end of 2021, admission numbers for all selected types of cancer surgeries reached pre-pandemic levels. The observed increase in prostatectomies in prostate cancer during spring lockdown

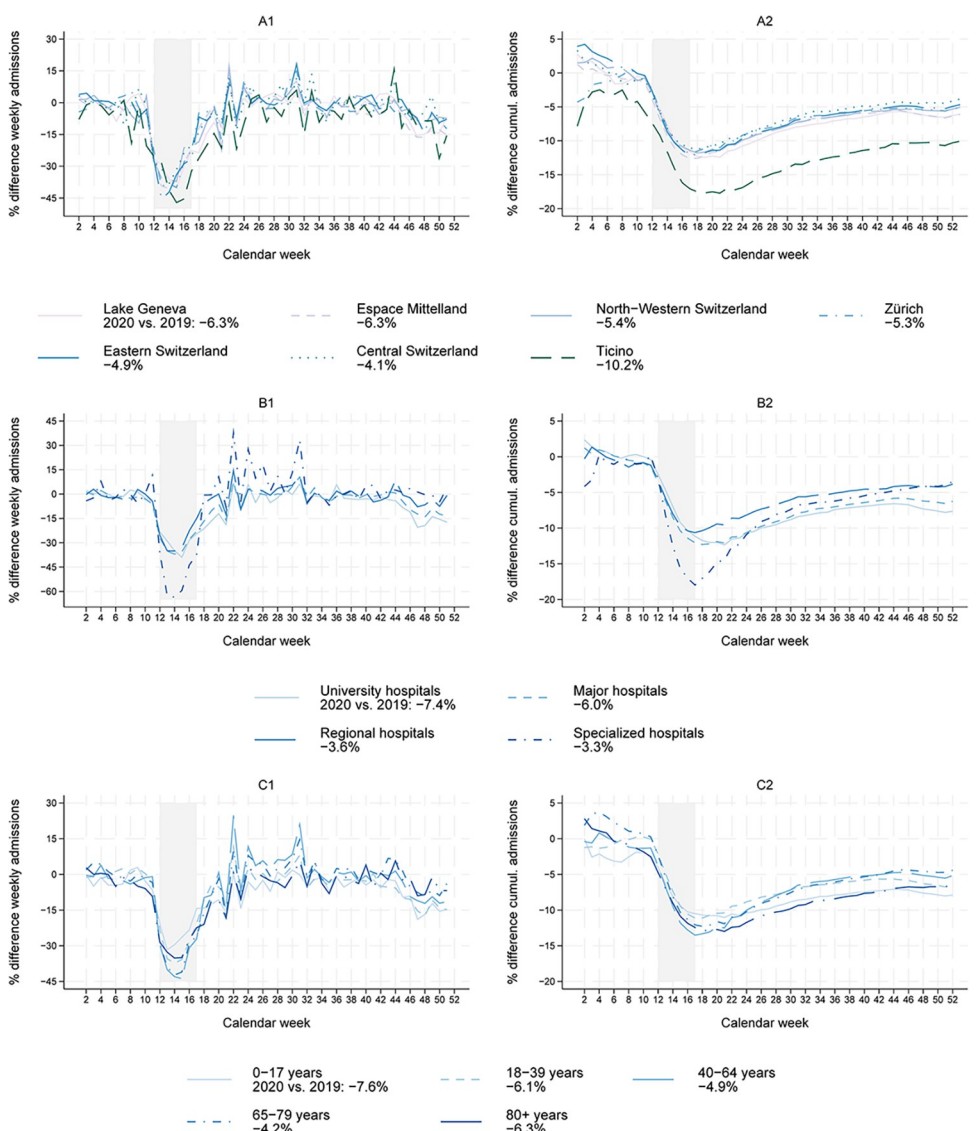

**Fig 2. Change in numbers of weekly admissions by regions, hospital type and age 2020 vs. 2019.** The graph shows the relative differences of the weekly admissions (1, left) and the cumulative differences (2, right) in 2020 compared to 2019 by regions (A, top), hospital type (B, middle) and patient age (C, bottom). The percentage value next to each subgroup label in the legend indicates the overall annual difference for that subgroup. Start is week 2 because this was the first complete week, i.e., the first week that consisted of seven days in the years 2019 and 2020. Week 52 is not displayed for the weekly admissions, as surgery activity in this week highly depends on when the holidays were within this week. The shaded area indicates the lockdown (calendar weeks 12–17).

and afterwards can be explained by a particularly low number of surgeries in the baseline year 2019 (compared to the previous years) (Fig 3, Table 2).

With a decline of between 8.7% and 26.3%, the number of the selected *emergencies* also substantially decreased during the spring lockdown: NSTEMI myocardial infarctions (no ST elevation on electrocardiogram (ECG), coronary angiography frequently indicated, depending on risk constellation [35, 36]) decreased by 26.3%, STEMI infarctions (ST elevation on ECG, immediate revascularization required [36]) by 8.7% (IRR not statistically significantly different from 1 with a 95%-CI of 0.8261–1.009), and appendectomy by 9.3%. In addition, 14.4% fewer

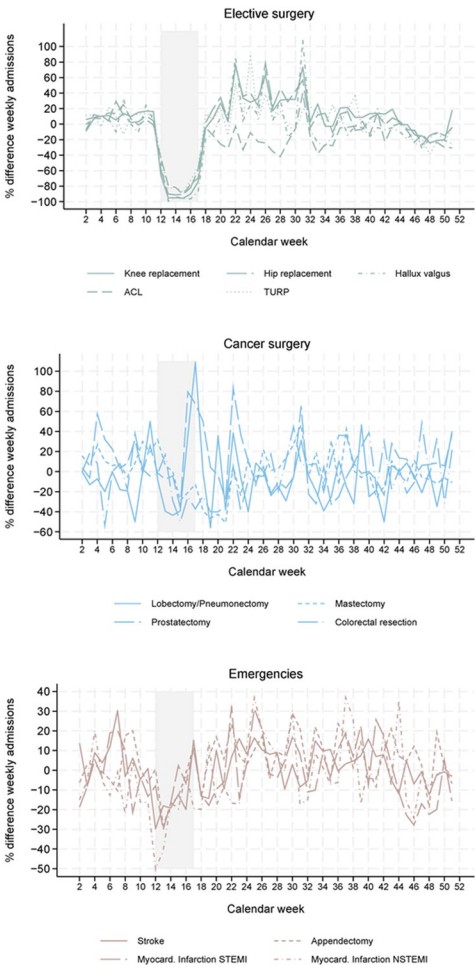

**Fig 3. Change in number of weekly admissions 2020 vs. 2019 for selected procedures.** Elective surgery (top row, left), cancer surgery (top row, right), emergencies (bottom row). Start is week 2 because this was the first complete week, i.e., the first week that consisted of seven days in the years 2019 and 2020. Week 52 is not displayed as surgery activity in this week highly depends on when the holidays were within this week. The shaded areas indicate the lockdown (calendar weeks 12–17).

patients were hospitalized with stroke. After the spring lockdown until the end of 2020, more patients were hospitalized for appendectomy, resulting in a relative increase of admissions by +4.4% by the end of 2020 compared to 2019. In contrast, still fewer people were hospitalized with an acute myocardial infarction of type NSTEMI after the spring lockdown compared to 2019 (the difference in absolute admission numbers further increased between spring lockdown and year-end 2020). Thus, all-year admission numbers remained at -4.6% slightly lower than in the previous year. The decrease was less pronounced for STEMI-type infarctions, as the statistically non-significant IRR indicates. In 2021, slightly more admissions were registered for stroke and appendectomy, while 5.1% less patients with NSTEMI were hospitalized compared to 2019 (Fig 3, Table 2).

## Discussion

Key findings of this study are that nationwide inpatient admission numbers decreased by 32% during the 2020 spring lockdown compared to the same period in 2019, and this reduction

**Table 2. Differences in admission numbers of selected procedures 2020–2021.**

| Procedure | Spring lockdown 2020 (weeks 12–17) | | | | | All-year 2020/2021 (weeks 2–52) | | | | | | |
|---|---|---|---|---|---|---|---|---|---|---|---|---|
| | 2019 | Difference 2019 vs. 2020 | | | 2019 | Difference 2019 vs. 2020 | | | Difference 2019 vs. 2021 | | | |
| | n | n | % | IRR (95%-CI) | n | n | % | IRR (95%-CI) | n | % | IRR (95%-CI) | |
| **Elective surgery** | | | | | | | | | | | | |
| Knee prosthesis (incl. partial prosthesis) for knee OA | 2151 | -1842 | -85.6 | 0.144 (0.128–0.162) | 18915 | -231 | -1.2 | 0.988 (0.968–1.008) | 1297 | 6.9 | 1.069 (1.048–1.090) | |
| Hip prosthesis (incl. partial prosthesis) for hip OA | 2050 | -1615 | -78.8 | 0.212 (0.191–0.235) | 18075 | -247 | -1.4 | 0.986 (0.966–1.007) | 1287 | 7.1 | 1.071 (1.050–1.093) | |
| Reconstruction of the anterior cruciate ligament | 1158 | -869 | -75.0 | 0.250 (0.219–0.284) | 6933 | -1404 | -20.3 | 0.797 (0.770–0.826) | -801 | -11.6 | 0.884 (0.855–0.915) | |
| Reconstruction of hallux valgus | 670 | -579 | -86.4 | 0.136 (0.109–0.169) | 5637 | -670 | -11.9 | 0.881 (0.848–0.915) | -860 | -15.3 | 0.847 (0.815–0.881) | |
| Transurethral prostatectomy (TUR-P) | 1026 | -733 | -71.4 | 0.286 (0.251–0.325) | 9054 | -399 | -4.4 | 0.956 (0.928–0.985) | 420 | 4.6 | 1.046 (1.017–1.077) | |
| Tonsillectomy (incl. adenoidectomy) | 888 | -717 | -80.7 | 0.193 (0.163–0.227) | 6882 | -1880 | -27.3 | 0.727 (0.701–0.754) | -2184 | -31.7 | 0.683 (0.658–0.708) | |
| **Cancer surgery** | | | | | | | | | | | | |
| Mastectomy in breast cancer | 778 | -36 | -4.6 | 0.954 (0.862–1.055) | 6464 | -42 | -0.6 | 0.994 (0.960–1.028) | 407 | 6.3 | 1.063 (1.027–1.100) | |
| Prostatectomy in prostate cancer | 348 | 26 | 7.5 | 1.075 (0.929–1.244) | 3172 | 151 | 4.8 | 1.048 (0.998–1.100) | 384 | 12.1 | 1.121 (1.069–1.176) | |
| Lobectomy or pneumonectomy in lung cancer | 128 | -17 | -13.3 | 0.867 (0.673–1.118) | 1085 | -130 | -12.0 | 0.880 (0.807–0.960) | -84 | -7.7 | 0.923 (0.847–1.005) | |
| Colorectal resection in colorectal cancer | 274 | -56 | -20.4 | 0.796 (0.666–0.951) | 2218 | -80 | -3.6 | 0.964 (0.908–1.023) | -9 | -0.4 | 0.996 (0.939–1.056) | |
| **Emergencies** | | | | | | | | | | | | |
| Stroke | 1568 | -226 | -14.4 | 0.856 (0.796–0.921) | 13695 | 149 | 1.1 | 1.011 (0.987–1.035) | 639 | 4.7 | 1.047 (1.022–1.071) | |
| Acute myocardial infarction NSTEMI | 1215 | -319 | -26.3 | 0.737 (0.676–0.804) | 9853 | -450 | -4.6 | 0.954 (0.928–0.982) | -504 | -5.1 | 0.949 (0.922–0.976) | |
| Acute myocardial infarction STEMI | 805 | -70 | -8.7 | 0.913 (0.826–1.009) | 7136 | -167 | -2.3 | 0.977 (0.945–1.009) | 160 | 2.2 | 1.022 (0.990–1.009) | |
| Appendectomy in acute appendicitis | 1297 | -120 | -9.3 | 0.907 (0.839–0.982) | 11017 | 483 | 4.4 | 1.044 (1.017–1.071) | 341 | 3.1 | 1.031 (1.004–1.058) | |

N = Number of admissions; OA = Osteoarthritis, IRR = Incidence Rate Ratio, calculated as $IRR = \frac{inpatient\ stays_{2020/2021}}{inpatient\ stays_{2019}}$.

was still visible at the end of the year. After the spring lockdown, admission numbers remained at a lower level than before the pandemic. As a result, Swiss hospitals treated 6% fewer inpatient cases in 2020 compared to 2019. Patterns subdivided by region, hospital type and age were generally similar, but the 2020 all-year reduction was highest in the South of Switzerland (canton Ticino) and for university hospitals. With up to 86%, the drop during spring lockdown was largest for elective surgery. Admissions in cancer surgery declined by up to 20% but varied considerably depending on cancer type. Also, emergencies such as heart attack and stroke dropped by up to 26%. The decline during spring lockdown was only fully reversed in three of the selected procedures (knee and hip protheses, TUR-P) by the end of 2020, while the numbers of most procedures remained by the end of 2020 at a lower level than before the pandemic. In 2021, numbers for most procedures again reached or even exceeded those in 2019. Only for some elective surgeries (reconstruction of hallux valgus and ACL) did admission numbers remain at a lower level than 2019.

The decrease in admissions of 32% during the spring lockdown shows that the federal triage in favor of anticipated Covid-19 cases had an effect. The number of reduced admissions is in line with results of an international systematic review that included data up to August 2020 and reported a median reduction in admissions of 28% (IQR -40 to -17%) in pandemic versus pre-pandemic periods for more than 20 countries all over the world [8]. In Switzerland, the impact of the Covid-19 pandemic on admissions was roughly similar in the different regions, with Ticino showing the largest reduction with the earliest onset. This can be explained by the geographical proximity and the close economic and social ties of Ticino with Lombardy, the first region in Europe to be severely affected by the Covid-19 pandemic as early as mid-February 2020 [37]. The fact that university hospitals and major hospitals presented the largest all-year deficits in admissions by the end of 2020 might reflect the fact that these hospitals contributed most to the management of the pandemic by treating large numbers of severely ill Covid-19 patients and by keeping intensive care unit capacities free in anticipation of more Covid-19 patients [38]. The lower numbers for middle-aged and elderly people (40–79 years) might be explained by the fact that this group tried to avoid using healthcare services during the spring lockdown. In line with this, a Dutch study found that mainly older adults reported having delayed seeking emergency healthcare, and the main reasons for this were fear of getting infected, official stay-at-home instructions and the intention to protect the healthcare system [39]. Also, particularly in lockdown periods, decreased mobility and restricted work and leisure activities led to a reduction in trauma patient admissions. For Switzerland, a decrease of 40 to 55% was noted during the spring 2020 lockdown [40], a trend that aligns with observations from other countries, including the US [41].

Among the selected procedures, there was a very large drop in the number of admissions for *elective surgery* during the spring lockdown, which was only fully reversed for total joint arthroplasty (and, approximately, for TUR-P) by the end of 2020. This catch-up effect for arthroplasty seems to be roughly comparable to the situation in the U.S., where the time to catch up the backlog of surgeries has been estimated to be three to twelve months [42]. In contrast, the catch-up effect was smaller in Germany [43]: 25% fewer hip replacements were carried out in the first half of 2020 (weeks 2–18) and still 20% fewer in the same period in 2021 than in 2019. Similarly, 23% fewer knee replacements were carried out in 2020 and 25% fewer in 2021 than in 2019 (7). Even before the pandemic, the waiting time for a hip prosthesis ranged from 39 days (Denmark) to 433 days (Chile) among the OECD-countries, and that for a knee prosthesis from 45 days (Denmark) to 861 days (Chile) [44]. The median waiting time for a hip replacement increased in 2020 by 58 days on average, and for a knee replacement by 88 days [24]. In Switzerland, there are hardly any waiting lists [45], even though it has the highest rates for knee replacements with 260 per 100,000 population (2019), and Switzerland is, with 313 hip replacements per 100,000, second only to Germany according to OECD figures [24]. The high capacities in this sector may favor overuse but allowed for a rapid catch-up of postponed procedures after the lockdown.

In *cancer surgery*, decreases in admissions during the spring lockdown varied, depending on cancer-type. The Federal Statistical Office reported a decrease in tumor-related hospitalizations of 16% during the spring lockdown and 4% by the end of 2020 compared to previous years (2017–2019) [46]. The observed large reduction in colorectal resection in this study is comparable to the numbers in German hospitals [10], while the increase in prostatectomies in prostate cancer is somewhat surprising but can be explained by the low number of surgeries in the baseline year 2019. The numbers of lobectomy/pneumonectomy in lung cancer clearly dropped during the spring lockdown and remained at a lower level afterwards. However, as this cancer is the least frequent type selected and its weekly admission numbers vary considerably, these figures need to be interpreted with caution. In addition, although we selected the

three most frequent types of cancer for men and women [30], respectively, the pattern found might not be representative for other cancer types. Lastly, based on our data we cannot assess what the observed surgical deferrals mean for individual patients, e.g., regarding the formation of metastases, and this requires long-term investigation.

There was also a large reduction in admissions for the selected *emergencies* during the spring lockdown, both for acute myocardial infarction and for cerebral stroke, which is consistent with results from the international literature [10, 15–17]. Although not conclusively clarified, this phenomenon is to some extent explained by the fact that patients either misinterpreted their symptoms because of the (media) focus on Covid-19 or did not go to hospital because of fears of a Covid-19 infection. It is also conceivable that certain pathologies occurred less frequently during the spring lockdown, e.g., because the population was exposed to fewer stressors or maintained a healthier lifestyle [15, 16]. The number of deaths due to cardiovascular diseases in 2020 increased by 2.3% for men and by 0.1% for women compared to 2019 [47], which is in the range of usual annual fluctuations [48, 49]. At the monthly level, there is no extraordinary increase in these causes of death during and after the spring lockdown compared with previous years [48, 49] or with January and February 2020 [50]. Thus, no more patients seem to have died from these life-threatening diseases compared with previous years despite the treatment ban. This is in line with an international meta-analysis that reported no effect of the lockdown on mortality for myocardial infarction [51]. However, the available data does not provide information on, for example, the quality of life and functional incapacities of affected patients. Thus, further studies are needed to investigate the actual impact of treatment bans and other restrictions, because, especially in the case of cerebral stroke, the resulting degree of disability is a crucial treatment outcome and is very much related to timely treatment onset. The fact that in acute myocardial infarction, the decline in hospitalizations during the spring lockdown was significantly greater for the (generally less severe) non-ST segment elevation myocardial infarction (NSTEMI) (-26%) compared to the ST segment elevation myocardial infarction (STEMI) (-9%) also confirms results from other countries: In German hospitals, the number of NSTEMI decreased by 18% between March and May 2020, that of STEMI by 9% [10]. A French registry study reported a decrease in NSTEMI by 35% and in STEMI by 24% between mid-February and mid-April [12], while in two Italian hub centers, NSTEMI admissions decreased by 35% and STEMI by 6% between early March and early April 2020 [52].

Altogether, the fact that admission numbers for most procedures remained at a lower level after spring lockdown compared to 2019 is somewhat surprising and the underlying reasons cannot be conclusively clarified based on our data. Since there was no longer a formal treatment ban in Switzerland, these reductions can probably be attributed to a reaction of the hospitals to the pandemic situation (higher needs for intensive care units [4], staff shortages due to covid-infections) and to changes in patients' demand for health services. Based on Swiss survey data, Höglinger et al. [5] found that both during and after the lockdown, numerous patients voluntarily refrained from seeking "planned or necessary" medical treatments "because of Corona". Similarly, a reduction in costs and consultations during the spring lockdown was observed in the Swiss outpatient sector [25, 53]. In qualitative interviews, patients specified "fear of infection", "following the public health messaging to stay at home", and a desire "not to further burden the overburdened general practitioners" as reasons for restricting their health services use [53]. In 2021, the total number of inpatient admissions almost reached the level of 2019, which was confirmed in a recent media release from the Federal Statistical Office [54]. However, admission numbers remained at a lower level compared to the pre-pandemic period in two of the selected procedures: reconstruction of hallux valgus and of ACL. This might reflect general trends in these procedures: hallux valgus surgery can be performed

in an outpatient setting [55], and in the treatment of a ruptured anterior cruciate ligament, the superiority of surgery over conservative treatment is controversial [56, 57].

## Strengths and limitations

The major strength of the present study is its completeness, as it is based on admission data of *all* Swiss hospitals. Thus, it comprehensively maps the processes in inpatient healthcare in Switzerland during the Covid-19 pandemic. The focus on selected procedures of different medical urgency from various medical fields allows for a detailed insight into the heterogenous impact of the pandemic in general, as well as of the treatment ban in particular, on inpatient procedures. On the other hand, our selection of procedures, although based on criteria and expert recommendations, is subjective to some extent. Another limitation is that we used 2019 as the baseline year to derive changes in admission numbers. The comparison with a single year might be hampered by random variations but, on the other hand, it minimizes any bias due to trends in disease prevalence, hospital infrastructure as well as changes in regulations and reimbursement schemes.

## Conclusion

The decline in admissions during the spring lockdown in Switzerland is similar to that reported in multinational studies [8]. Admission numbers did not return to pre-pandemic levels by the end of 2020, but did so, with few exceptions, by the end of 2021. With up to 86%, the drop during spring lockdown was largest for elective surgery, but also emergencies such as heart attack and stroke and some selected cancer surgeries were affected. This decline was generally not fully reversed by the end of 2020 except for three of the selected elective procedures (knee and hip protheses, TUR-P). After the spring lockdown until the end of 2020, most procedures remained at a lower level than before the pandemic. Only in 2021 did most admission numbers reach or exceed those of 2019, which suggests that the Covid-19 pandemic did not, in general, have a lasting impact on inpatient health service use in Switzerland. The findings show the Swiss healthcare system's ability to swiftly adapt to the pandemic's demands. This resilience, often scrutinized for its perceived excess resources and high costs, highlights the critical role of resource availability in the system's effective response to the crisis. However, the long-term consequences of the observed reductions in admissions for emergencies and cancer surgery for patients need to be investigated.

## Acknowledgments

We thank Simon Wieser, Klaus Eichler, Marcel Widmer and Olivier Pahud for their valuable input. We also thank the experts who we interviewed for sharing their expertise, the coding specialists for their support in assigning the correct codes to the selected procedures, and Paul Kelly for editing the manuscript. We are grateful to the three reviewers for their insightful suggestions to enhance our manuscript.

## Author Contributions

**Conceptualization:** Brigitte Wirth, Michael Stucki, Reto Joerg, Christoph Thommen, Marc Höglinger.

**Data curation:** Reto Joerg.

**Formal analysis:** Michael Stucki, Reto Joerg.

**Funding acquisition:** Marc Höglinger.

**Investigation:** Brigitte Wirth, Michael Stucki, Christoph Thommen, Marc Höglinger.

**Methodology:** Brigitte Wirth, Michael Stucki, Reto Joerg, Christoph Thommen, Marc Höglinger.

**Visualization:** Michael Stucki.

**Writing – original draft:** Brigitte Wirth, Christoph Thommen.

**Writing – review & editing:** Brigitte Wirth, Michael Stucki, Reto Joerg, Marc Höglinger.

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
