## [Decision Letter · Decision Letter 0]

19 Mar 2024

PONE-D-23-31239Impact of the Covid-19 pandemic on inpatient health care in Switzerland 2020-2021 – a retrospective study using admission data of all Swiss hospitalsPLOS ONE

Dear Dr. Höglinger,

Thank you for submitting your manuscript to PLOS ONE. After careful consideration, we feel that it has merit but does not fully meet PLOS ONE’s publication criteria as it currently stands. Therefore, we invite you to submit a revised version of the manuscript that addresses the points raised during the review process.

We look forward to receiving your revised manuscript.

Kind regards,

Francesco Sessa, Ph.D., MS

Academic Editor

PLOS ONE

Journal Requirements:

"The study was funded by the Swiss Health Observatory (OBSAN)"

3. For studies involving third-party data, we encourage authors to share any data specific to their analyses that they can legally distribute. PLOS recognizes, however, that authors may be using third-party data they do not have the rights to share. When third-party data cannot be publicly shared, authors must provide all information necessary for interested researchers to apply to gain access to the data. (https://journals.plos.org/plosone/s/data-availability#loc-acceptable-data-access-restrictions) 

Additional Editor Comments:

The reviewers raised several important concerns about the manuscript: please, provide a point-by-point response in a rebuttal letter highlighting all changes made in the main text.

Reviewers' comments:

Reviewer's Responses to Questions

**Comments to the Author**

1. Is the manuscript technically sound, and do the data support the conclusions?

Reviewer #1: Partly

Reviewer #2: Yes

Reviewer #3: Yes

2. Has the statistical analysis been performed appropriately and rigorously? 

Reviewer #1: No

Reviewer #2: Yes

Reviewer #3: Yes

3. Have the authors made all data underlying the findings in their manuscript fully available?

Reviewer #1: No

Reviewer #2: Yes

Reviewer #3: Yes

4. Is the manuscript presented in an intelligible fashion and written in standard English?

Reviewer #1: Yes

Reviewer #2: Yes

Reviewer #3: Yes

5. Review Comments to the Author

Reviewer #1: Comments: In this study, the authors described the changes of overall and procedure-specific hospital admissions in Switzerland during the COVID-19 pandemic in 2020 and 2021.

Main concern:

The authors estimated the changes in hospital admissions by directly comparing the differences between periods during the pandemic and corresponding periods in 2019. The problem with this method is that it couldn’t account for the long-term trend of hospital admissions. Although the authors explained in the methods, the percentage changes in 2020 and 2021 were small (5.8% and 0.9% less than in 2019), there is the possibility that the reduction was attributed to the long-term reducing trend rather than the effects of the pandemic. I can see from Figure 1 that the authors had the data during 2016 and 2021. Therefore, I would like to suggest predicting the weekly number of hospitalizations in 2020 and 2021 using time-series data from 2016-2019 by fitting Poisson or Negative Binomial Distribution Models, accounting for the long-term trend and seasonality. Then to compare the difference between observed and predicted values in the study period to estimate the absolute or percentage changes. This method could also solve the uncertainties issue, which is a problem in this study. If the authors insist on using the current method (comparing study periods with corresponding periods in 2019), I would like to suggest another method used in the Filippo et al._ J Med_2020 paper, which would help to solve the uncertainties issue.

Minor concern

In figures 2 and 3, I would suggest labeling each part (for example, “A”, “B”, “C”, “D”), and to show us the overall percentage changes for each group rather than the weekly changes.

Reviewer #2: The article appears well written and has a good research methodology which has been clearly stated. The topic is certainly interesting and useful not only for observation purposes but also for research purposes and interest in the planning of health systems in the Western world, I therefore believe that the article should be published.

I have some suggestions:

- the authors examine the results of their analysis which appear globally comparable to other international data and try to give explanations regarding them. I have not seen that a fundamental aspect for the healthcare systems of European countries, namely the increase in the average age, is taken into consideration. The authors could mention this variable indicating whether this had any interaction in the study.

- the results show a decline in access to the emergency room/emergency during the lockdown period but also subsequently; the fact that following the lockdown but also in the subsequent period there were restrictions on the traffic of people or in any case there was less traffic on the road, this aspect may have reduced car accidents and therefore access for this reason. Could this be a reason to add and discuss briefly?

- the conclusions are a bit lacking. Could some useful considerations be made for designing the future of healthcare systems to make them more resilient?

Reviewer #3: The article is well structured however it does not provide any innovative elements. Substantially, it is a descriptive analysis of three years 2019, 2020, 2021 of hospitalizations but provides only a modest scientific contribution. it is, in fact, logical that, during the lockdown there was a decline in hospitalizations with a subsequent surge, this is a fact already widely described in the literature. It would have been useful, however, to find out whether the reduction in hospitalizations has led, for example, to a delay in the diagnosis of some cancers. In summary, I do not believe that the scientific contribution of this manuscript is useful today.

6. PLOS authors have the option to publish the peer review history of their article (what does this mean?). If published, this will include your full peer review and any attached files.

Reviewer #1: No

Reviewer #2: **Yes: **Matteo Bolcato

Reviewer #3: No

---

## [Author Response · Author response to Decision Letter 0]

28 Apr 2024

Please see the "Response to reviewers" document.

---

## [Decision Letter · Decision Letter 1]

25 Jun 2024

Impact of the Covid-19 pandemic on inpatient health care in Switzerland 2020-2021 – a descriptive retrospective study using admission data of all Swiss hospitals

PONE-D-23-31239R1

Dear Dr. Höglinger,

We’re pleased to inform you that your manuscript has been judged scientifically suitable for publication and will be formally accepted for publication once it meets all outstanding technical requirements.

Kind regards,

Francesco Sessa, Ph.D., MS

Academic Editor

PLOS ONE

Additional Editor Comments (optional):

The authors modified the manuscript following the reviewers' suggestions.

Reviewers' comments:

Reviewer's Responses to Questions

**Comments to the Author**

1. If the authors have adequately addressed your comments raised in a previous round of review and you feel that this manuscript is now acceptable for publication, you may indicate that here to bypass the “Comments to the Author” section, enter your conflict of interest statement in the “Confidential to Editor” section, and submit your "Accept" recommendation.

Reviewer #2: (No Response)

2. Is the manuscript technically sound, and do the data support the conclusions?

Reviewer #2: (No Response)

3. Has the statistical analysis been performed appropriately and rigorously? 

Reviewer #2: (No Response)

4. Have the authors made all data underlying the findings in their manuscript fully available?

Reviewer #2: (No Response)

5. Is the manuscript presented in an intelligible fashion and written in standard English?

Reviewer #2: (No Response)

6. Review Comments to the Author

Reviewer #2: the authors have improved the text according to the suggestions that were previously indicated. The article in my opinion can now be published

7. PLOS authors have the option to publish the peer review history of their article (what does this mean?). If published, this will include your full peer review and any attached files.

Reviewer #2: **Yes: **Matteo Bolcato

---

## [Editor Report · Acceptance letter]

27 Jun 2024

PONE-D-23-31239R1 

PLOS ONE

Dear Dr. Höglinger, 

I'm pleased to inform you that your manuscript has been deemed suitable for publication in PLOS ONE. Congratulations! Your manuscript is now being handed over to our production team.

Kind regards, 

on behalf of

Lecturer Francesco Sessa 

Academic Editor

PLOS ONE